# The HIV-1 Integrase C-Terminal Domain Induces TAR RNA Structural Changes Promoting Tat Binding

**DOI:** 10.3390/ijms232213742

**Published:** 2022-11-08

**Authors:** Cecilia Rocchi, Camille Louvat, Adriana Erica Miele, Julien Batisse, Christophe Guillon, Lionel Ballut, Daniela Lener, Matteo Negroni, Marc Ruff, Patrice Gouet, Francesca Fiorini

**Affiliations:** 1Molecular Microbiology and Structural Biochemistry, MMSB-IBCP, UMR 5086, CNRS, University of Lyon, 7 Passage du Vercors, CEDEX 07, 69367 Lyon, France; 2Institute of Analytical Sciences, UMR 5280 CNRS UCBL University of Lyon, 5 Rue de la Doua, 69100 Villeurbanne, France; 3Department of Biochemical Sciences, Sapienza University of Rome, P.le Aldo Moro 5, 00185 Rome, Italy; 4Chromatin Stability and DNA Mobility, Department of Integrated Structural Biology, IGBMC, CNRS, UMR 7104—Inserm U 158, University of Strasbourg, 1 rue Laurent Fries, 67404 Illkirch, France; 5RNA Architecture and Reactivity, IBMC, CNRS, UPR 9002, University of Strasbourg, 2, Allée Konrad Roentgen, 67084 Strasbourg, France

**Keywords:** RNA-protein interaction, HIV-1 Integrase, TAR RNA, Tat, proviral transcription, C-terminal tail

## Abstract

Recent evidence indicates that the HIV-1 Integrase (IN) binds the viral genomic RNA (gRNA), playing a critical role in the morphogenesis of the viral particle and in the stability of the gRNA once in the host cell. By combining biophysical, molecular biology, and biochemical approaches, we found that the 18-residues flexible C-terminal tail of IN acts as a sensor of the peculiar apical structure of the trans-activation response element RNA (TAR), interacting with its hexaloop. We show that the binding of the whole IN C-terminal domain modifies TAR structure, exposing critical nucleotides. These modifications favour the subsequent binding of the HIV transcriptional trans-activator Tat to TAR, finally displacing IN from TAR. Based on these results, we propose that IN assists the binding of Tat to TAR RNA. This working model provides a mechanistic sketch accounting for the emerging role of IN in the early stages of proviral transcription and could help in the design of anti-HIV-1 therapeutics against this new target of the viral infectious cycle.

## 1. Introduction

Protein–nucleic acid interactions can occur through different types of protein binding domains and are responsible for a variety of essential molecular and cellular mechanisms and their regulation. This binding diversity has been well described by recent RNA interactome screenings revealing that the term RNA-Binding Domain (RBD) is no longer synonymous of a well-structured domain, but it also includes intrinsically disordered regions (IDR) endowed with non-canonical RNA-binding properties [1,2,3]. Moreover, an increasing level of complexity has been reported for certain transcription factors, which are able to bind both DNA and RNA through separate structured or unstructured regions. This results in a complex pattern of specific and non-specific interactions [4,5]. This protein moonlighting is particularly true for RNA viruses that, being limited in the size of their genome, frequently ensure multiple functions with a single protein [6,7].

A good example of moonlighting is represented by Human Immunodeficiency Virus type 1 (HIV-1) integrase (IN) that binds both DNA and RNA [8,9]. As with all retroviruses, IN is responsible for the integration of viral DNA (vDNA), which is produced by retro-transcription of genomic RNA (gRNA) into the host genome, generating a provirus. Multimeric IN binds the vDNA ends forming the intasome complex able to catalyze the processing of the 3′-end di- or trinucleotides. After activation, the 3′ extremities of vDNA are then used by the intasome to attack the host DNA in order to integrate it and generate the provirus (reviewed in [9]). This step is essential for HIV-1 infection [10]. Retroviral integrases are modular proteins that contain three structured domains: the N-terminal domain (NTD), the catalytic core domain (CCD), and the C-terminal domain (CTD), connected by linkers, either unstructured or helical [11]. All three domains present protein–protein and protein–DNA interaction properties and are essential for enzymatic activity (reviewed in [9,12]). The CCD harbors the essential catalytic triad Asp, Asp, Glu (D,D,E) that coordinates two Mg^2+^ cofactors and folds similarly to nucleotidyl-transferases and nucleases [13]. The NTD is involved in enzyme multimerization and catalytic activity and shows a His_2_Cys_2_ motif coordinating one Zn^2+^ ion [14,15,16]. The CTD (reviewed in [11]) is also involved in DNA interaction and multimerization, and it possesses an SH3-like fold followed by a flexible 18-residues tail (CT) [17,18,19,20]. The SH3-like domain of IN-CTD shows non-specific DNA-binding properties and contains many basic residues [18,20,21,22]. In the HIV-1 life cycle, this hub domain mediates the interaction with the reverse transcriptase (RT) with the cellular nuclear import complex TRN-SR2 and with the histone 4 tail, likely anchoring the intasome to the chromatin, thereby promoting efficient integration [21,23,24,25]. The mutational study of flexible CT revealed a moderate implication in IN enzymatic activity [22,23,26] but significant effect on the incorporation of IN into virions and on HIV-1 infectivity [23,26]. Nevertheless, our understanding of the multiple functions exerted by this domain during infection is still far from complete.

As mentioned before, recent works revealed that in HIV-1, IN is also an RBP with an essential role in virion morphogenesis related to its ability to bind gRNA [8,27,28]. In fact, IN interacts with specific sites within the gRNA, ensuring the correct localization of the viral ribonucleopotein complex (vRNP) inside the capsid (reviewed in [27]). Aberrant virions are obtained when the IN–gRNA interaction is abolished, highlighting the importance of the proper formation of IN-containing RNPs for HIV-1 infection [8,26,28,29]. In addition, the IN–gRNA interaction also dictates the fate of the gRNA within the host cell in the early steps of infection. Indeed, in virions defective for these interactions, the gRNA is located outside the capsid and it is rapidly degraded in the host cell, blocking the infectious cycle at early stages of reverse transcription [28,29]. Even if the RBD within IN has not yet been structurally identified, it has been shown that most of the Lysine residues interacting with the RNA are located within the CTD and overlap with those subjected to post-translational modification [8,30,31,32]. Among these, Lysine 273, belonging to the flexible CT, seems to be the only Lysine specifically involved in RNA interaction, which appears essential for viral infectivity [8,33]. Several compensatory IN mutant viruses have been recently isolated and highlighted the essential role of CTD in RNA binding [34].

Yet another role has been recently proposed for HIV-1 IN during proviral transcription at early times after integration. In fact, after strand transfer occurring during the integration process, IN remains bound to DNA and plays a role in proviral transcription, depending on post-translational modifications of specific residues within the CTD [35].

The HIV-1 provirus is transcribed by the cellular RNA polymerase II (Pol II), which pauses shortly after the initiation of transcription due to the presence of negative elongation factors as well as nucleosomes downstream from the transcription start site [36,37]. HIV-1 removes this block by encoding the transcriptional trans-activator Tat protein, which binds the nascent transcript on a structured RNA sequence, named the TAR (trans-activation response) element, using a non-canonical RBD. This allows the recruitment of the human super elongation complex (hSEC) [38,39,40,41]. In particular, Tat binds to pTEFb, a complex composed of CDK9 kinase and its regulatory partner cyclin T1 (CycT1), and consequently drives hSEC to TAR RNA. This complex triggers a cascade of phosphorylation of several transcription factors, which activate Pol II and recruit positive chromatin remodellers. Moreover, processivity of Pol II is also enhanced by pTEFb-mediated phosphorylation of its C-terminal domain [42] and is reviewed in [43]. To the best of our knowledge, molecular details about the interplay between IN and this cellular transcription initiation machinery are still largely unknown.

Crosslinking-immuno-precipitation sequencing (CLIP-seq) experiments have identified the TAR RNA sequence as a major binding site of HIV-1 IN on the gRNA [8]. This observation, together with the recent finding of HIV-1 IN involvement in proviral transcription [35], prompted us to study the interaction of IN with TAR RNA and its interplay with the Tat protein. Our results revealed that, despite the apparent lack of structural specificity of IN in vitro, the CT flexible tail discriminates for a proper TAR apical stem-loop. We describe the consequences of IN binding on the structure of TAR and on the subsequent Tat/TAR interaction and propose a working model, which foresees a possible involvement of IN in proviral transcription elongation before the arrival of Tat.

## 2. Results

### 2.1. IN Binds TAR RNA with No Apparent Structural Specificity

We first addressed whether full-length HIV-1 IN (IN-FL) was able to specifically bind TAR RNA. One hindering aspect of this study is the well-known low solubility of HIV-1 IN as well as its flexibility between N-terminal (NTD), C-terminal (CTD), and catalytic core (CCD) domains that for a long time frustrated structural studies (Figure 1A [44]). The poor solubility in vitro is usually overcome with mutations of the hydrophobic residues; however, this results in a replication-defective virus with mislocalized viral RNP phenotype, analogous to that observed in IN mutants defective for the IN–gRNA interactions [8,19,28,29]. We have chosen to express recombinant wild-type N-Terminal Flag-tagged IN-FL in a mammalian expression system as previously described ([45] Appendix A) and called it IN-FLm (1-288 amino acids [aa]). Mass spectrometry analysis of this protein revealed post-translational modification at several residues; Serine 24 was phosphorylated, and Lysine 46, 173, 211, and 273 residues were acetylated. The protein produced in mammalian cells was shown to have an increased solubility compared to that produced in *Escherichia coli* and also an enhanced enzymatic activity in vitro [45]. In this study we used an electrophoretic mobility shift assay (EMSA) in order to qualitatively assess the binding of IN to a synthetic TAR RNA, whose secondary structure is depicted in Figure 1B, to a weakly-structured RNA(30)-mer (Appendix A) and to an unstructured AG(50)-mer RNA (Figure 1C). We have incubated radiolabeled RNA with increasing concentrations of purified IN-FLm. A band shift was observed with TAR RNA upon gel electrophoresis under a nearly physiological salt concentration, reflecting the formation of an IN:TAR complex (Figure 1C). IN-FLm also bound weakly-structured RNA(30)-mer (Appendix A) while it had a markedly reduced affinity for unstructured AG(50)-mer RNA (Figure 1C). IN-FL produced in the prokaryotic expression system (Appendix A) is also able to bind TAR (Appendix A). Therefore, the post-translational modifications are not key to binding.

As RNA-interacting Lysine residues are contained within the C-terminal domain of IN [8], and given the premises stated in the introduction, we focused on this domain. We expressed and purified the whole C-Terminal Domain (IN-CTD; aa 220–288) and determined whether IN-CTD was able to interact with structurally distinct viral genomic RNA elements of similar nucleotide length derived from HIV-1 5′ UTR. In addition to TAR, we synthesized three other RNA hairpins: the polyadenylation (polyA) signal, the dimerization initiation sequence (DIS), and the major splice-donor (SD) together with a Ψ packaging element (Psi) (Figure 1D, top panel). Although different K_D_ values were measured using IN-FL [8], IN-CTD bound all those RNAs with no measurable difference in our conditions (Figure 1D, bottom panel), indirectly proving that IN can bind structured RNA through its C-terminal domain. 

Subsequently, we wanted to address whether the C-terminal flexible tail (CT) spanning the last 18 residues of CTD affected the RNA binding properties of IN. We expressed the CTD without its terminal tail (IN-CTD-ΔCT; aa 220–270, Figure 1A and Appendix A) and used a chemically synthesized IN-CT peptide (aa 270–288; Figure 1A) as a control. The boundaries between CTD and CT were defined according to sequence alignment and previous structural studies [17,33,46,47]. The IN-CT is not conserved among lentiviruses (Appendix A); however, multiple alignments of IN-CT from human HIV-1 subtypes and simian viruses (Figure 1E) showed 58% sequence identity and 87% similarity. EMSA assays did not show apparent differences between the affinities of IN-CTD and IN-CTD-ΔCT for TAR under physiological salt conditions (Appendix A). To measure the rate constants of the IN-CTD and IN-CTD-ΔCT interactions with TAR, we used Bio-layer interferometry (BLI). In these experiments, a 3′ biotinylated TAR RNA was immobilized on a streptavidin-coated biosensor to serve as a bait molecule (Figure 1F, top panel). Subsequently, the binding of IN subdomains to TAR was monitored in real time by soaking the probe in solutions containing different concentrations of protein. The resulting sensorgrams are shown in Appendix A and represent the binding and dissociation of the protein on the RNA-coated probe. The data analysis resulted in equilibrium dissociation constant (K_D_) values of 770 and 320 nM for IN-CTD-ΔCT and IN-CTD, respectively (Figure 1F, bottom panel). Thus, the presence of the 18 aa C-terminal tail only slightly affects the RNA binding affinity of IN-CTD.

**Figure 1 ijms-23-13742-f001:**
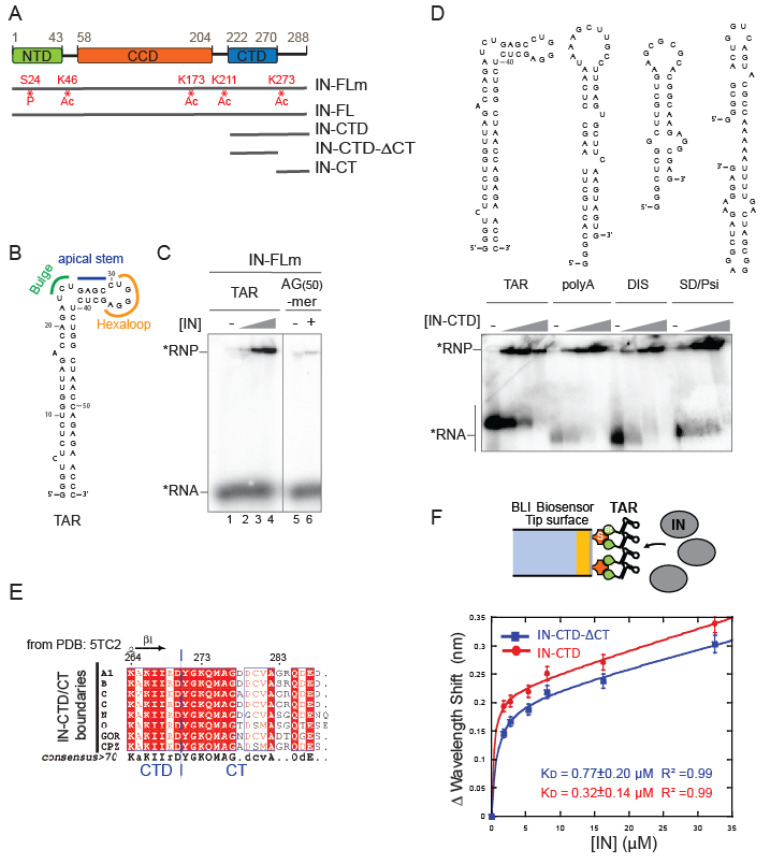
IN binds to structured RNA. (**A**) Schematic diagram showing the domain organization of IN: the N-terminal domain (NTD), the catalytic core domain (CCD) and the C-terminal domain (CTD) are indicated respectively in green, orange and blue rectangles. Recombinant protein versions used in this study are represented by grey lines: IN full-length from mammalian expression system (IN-FLm); IN full-length from *E. coli* (IN-FL); IN-CTD (residues 222 to 288); IN-CTD-ΔCT (222 to 270); IN-CT (270 to 288). Sites of phosphorylation (P) and acetylation (Ac) are shown in red. (**B**) Structural model of TAR RNA obtained with RNA Fold WebServer [48,49] predicting a minimum free energy of −29.60 kcal/mol. (**C**) Representative native 6% polyacrylamide gel illustrating the Electrophoretic mobility shift assay (EMSA) showing the interaction of IN-FLm with TAR RNA or an unstructured RNA 50-mer (AG(50)-mer) labelled with 32P (black star). The RNA substrates (50 nM) were incubated with various concentrations of IN-FLm or without protein (TAR RNA: 0; 100, 200, 400 nM of IN-FLm; AG(50)-mer RNA: 0 or 400 nM of IN-FLm). (**D**) Models of secondary structures of four RNA elements belonging to 5′ untranslated region of the HIV-1 RNA genome: TAR, Poly-A, DIS and SD/Psi (upper panel). EMSA assay showing the binding of IN-CTD to these RNA elements (lower panel). Increasing amounts of IN-CTD (0; 100; 200; 400 nM) were incubated with 5′-end radiolabeled RNA (50 nM). (**E**) Sequence alignment of the C-terminal extremity of IN-CTD from HIV-1 subtypes and simian viruses. All amino acid sequences were obtained from the HIV database compendium (http://www.hiv.lanl.gov/, accessed on 28 October 2020) and aligned using Clustal Omega (https://www.ebi.ac.uk/Tools/msa/clustalo/, accessed on 28 October 2020) in order to have a consensus sequence for each viral subtype. Consensus sequences from subtypes A1, B, C, G, N, O and from GOR and CPZ, where aligned and analyzed by ESPript 3.0 Web server [50]. Secondary structure elements from IN-CTD- ΔCT structure (PDB code: 5TC2) are presented on top of the alignment (strands with arrows). Red shading indicates sequence identity and boxes indicate sequence similarity, according to physical-chemical properties. (**F**) top: Schematic representation of the Bio-Layer interferometry (BLI) experiment showing the binding of IN (grey ellipses) to 5′-biotinylated TAR RNA immobilized on streptavidin-coated biosensor; bottom: graph showing the wavelength shifts recorded at 200 s after the start of the protein/RNA binding were plotted against the corresponding IN-CTD-ΔCT (blue line, squares) and IN-CTD (red line, circles) concentrations, in order to calculate the respective equilibrium dissociation constant (K_D_) values. Data points were fitted to the equation: y = Bmax × x/(K_D_ + x) + NS × x where Bmax is the maximum wavelength shift and NS the slope of the non-linear component as described in [51]. The coefficients of determination (R2) and equilibrium dissociation constant (K_D_) values obtained are indicated in the graph for each IN fragment. Binding assays were performed in duplicate. Error bars indicate the Standard Error of the Mean.

### 2.2. The C-Terminal Tail Senses the TAR RNA Shape

It has been shown that the presence of the bulge and the loop in TAR RNA, rather than its sequence, is critical for IN binding, as the deletion of one or both markedly decreased IN binding affinity [8]. We wondered whether the shape of the peculiar apical stem-loop of TAR (Figure 2A) could also be important. Therefore, we mutated the 4 nucleotides (nt) stem between the bulge and the loop to change its length, as shortening or lengthening the 4 nt-stem is likely to tighten or loosen the TAR major groove by bringing the bulge and the hexa-loop closer or further, respectively. We produced one TAR with 1 bp longer stem (TAR-LS), two with progressively shorter stem TARs (TAR-SS and TAR-VSS) (Figure 2A), and one without a stem (TAR-ΔS) (Appendix A). The EMSA assay showed that IN-FLm was able to bind TAR-LS similarly to the wild-type TAR (Appendix A). On the contrary, shortening of the stem reduced IN-FLm affinity for TAR (Appendix A). Furthermore, we assessed the binding behavior of IN-CTD and IN-CTD-ΔCT to TAR mutants with EMSA (Figure 2B). As observed for the full-length protein, the affinity of IN-CTD for TAR mutants decreased as the length of the stem was reduced (Figure 2B, lanes A to D). Surprisingly, the binding ability of IN-CTD-ΔCT was not affected by the shortening of the TAR 4nt-stem (Figure 2B, lanes A’ to D’), suggesting that IN C-terminal tail senses the shape of the TAR RNA stem in vitro.

As a control, we studied the interaction of Tat with TAR, the cognate interacting protein. We used a chemically synthesized full-length Tat protein (1-102 aa) to assess the binding of the TAR mutants with an EMSA assay. Similarly to IN-CTD and consistently with other in vitro and ex vivo information [52,53,54,55], the binding ability of Tat decreased with stem shortening (Appendix A).

Overall, our results suggest that IN-CTD interacts with the apical stem-loop of TAR, possibly with its major groove. This is consistent with the CLIP-seq data, which had identified the TAR hexaloop as a major binding site for IN, and with a recently published structural model of IN-CTD-ΔCT bound with TAR [8,56]. This is also consistent with cross-linking SHAPE experiments published while this manuscript was under submission [57].

### 2.3. IN-CTD Deeply Affects the Structure of TAR Favoring Tat Interaction

In order to extend the binding analysis at the molecular level, the IN-TAR interaction was probed by footprinting techniques. The 5′ radiolabelled TAR, alone or complexed to a protein, was subjected to partial digestion using RNase T1 that cleaves the 3′ to an unpaired guanine. As was consistent with previous results [58], under our conditions Tat- bound TAR RNA formed a large complex and prevented the RNA digestion by the nuclease (Figure 3A, lane 7). Concerning the interaction with IN fragments, the results shown in Figure 3A indicated that G34 and, to a lesser extent, G36 of TAR were protected in the presence of IN-CTD (Figure 3A, lane 6). However, the binding of IN-CT and IN-CTD-ΔCT did not protect these nucleotides from nuclease digestion (Figure 3A, lanes 4 and 5). This suggested that the CT region is interacting with G34 and possibly with G36 located in the TAR hexaloop and its junction with the 4-nt stem, but only when CT is part of the whole domain (Figure 3A, lanes 4 to 6). To our surprise, we observed a prominent cleavage following the residues at positions C41, U38, G32, and, to a lesser extent, G28, which was induced by the binding of IN-CTD-ΔCT and IN-CTD even in the absence of a T1 nuclease (Appendix A). This could reflect the presence of drastic structural constraints on the apical stem-loop of TAR upon the binding of either IN-CTD-ΔCT or IN-CTD, which leads to a spontaneous mechanical breakage/hydrolysis in the frame of our small TAR RNA construct. No nuclease contamination has been observed in the protein solution (Appendix A), and the data was reproducible.

Altogether, the digestion patterns showing the protection of G34 and G36 by IN-CTD, but not IN-CTD-ΔCT, as well as the evidence of the structural deformation of TAR induced by the binding of both fragments, indicate that (i) IN-CTD binds to the TAR apical stem-loop, (ii) IN-CTD modifies the structure of the RNA, (iii) and the CT exerts a specific role in the interaction of IN-CTD with the TAR hexaloop and its junction with the stem.

The fact that CT binds TAR only in the IN-CTD context could reflect the necessity for a distortion of TAR by the IN-CTD-ΔCT moiety to allow for the correct binding of CT to the hexaloop. Interestingly, in a recent work, Liu and co-workers [57] studied the binding of IN-FL to a whole HIV-1 5′UTR with crosslinking followed by SHAPE analysis. They found the crosslinking site for IN-FL near our breakage site (C41, U38) at position C39. Moreover, they also found an increased SHAPE reactivity of G32 indicating its greater exposure upon IN-FL binding [57] as we could show for IN-CTD.

In order to understand whether the structural changes of TAR induced by IN binding were affecting the affinity of Tat to TAR, we performed real-time binding experiments with Bio-Layer Interferometry (BLI). We measured the interactions as follows (Figure 3B, upper scheme). First, we immobilized biotinylated TAR RNA to a streptavidin-coated biosensor and washed out the unbound RNA (Figure 3B, curve between 50 and 170 s and between 170 s and 200 s); then, we soaked the probe in solutions containing a large excess (more than 10 times the K_D_) of IN-CTD or IN-CTD-ΔCT (Figure 3B, ascendant curve between 200 and 400 s indicating positive interference and hence association). Afterwards, the IN:TAR bound-biosensor was plunged in a solution containing Tat. Different concentrations of Tat were analyzed each time on different probes prepared as described in the methods. According to the experimental results in Figure 3B, a supplementary association of Tat on the IN:TAR complex occurred as indicated by the second ascendant curve between 400 and 600 s (Figure 3B). Interestingly, this association was dose-dependent as demonstrated by using increasing concentrations of Tat, between 0.2 and 2 µM (Figure 3B).

In order to quantify this observation, we calculated the apparent K_D_ of Tat upon the binding of IN (Figure 3C) and found that it was about 34.7 and 114.6 times lower in presence of IN-CTD-ΔCT and IN-CTD, respectively (Figure 3C), compared to the K_D_ of Tat measured in the same conditions for the naked RNA (Figure 3D and Appendix A). Moreover, the binding kinetics of Tat on the IN:TAR complex is consistent with a cooperative interaction (Figure 3C). Notably, the IN-CTD:TAR complex can accommodate more Tat than IN-CTD-ΔCT:TAR, as indicated by the higher BMax (maximum wavelength shift, Figure 3C). We performed the same BLI experiments coating the biosensor with the biotinylated generic RNA(30)-mer used in the EMSA assay (Appendix A) instead of TAR RNA, and we repeated the same protocol (Appendix A). Indeed, there is a displacement, but, comparing the K_D_ of Tat binding to naked RNA(30)-mer (Figure 3D) to the apparent KD after the binding of IN (Figure 3E), we observed a decrease of only about 6 times. Moreover, the experimental points followed a hyperbolic Michaelis–Menten kinetics (Figure 3E) different from the cooperative sigmoidal Hill kinetics observed for TAR (Figure 3C). Therefore, CT within IN-CTD enhances Tat binding to TAR and makes the binding cooperative.

To prove that these results were not the consequence of a direct interaction between IN-CTD and Tat, we performed a pulldown assay, which did not show protein–protein interaction in these conditions (Appendix A).

We also performed the competition experiment by reversing the order of addition of proteins (Appendix A); after TAR or RNA(30)-mer loading and wash, we first incubated the probe with Tat (Appendix A ascendant curve between 200 and 400 s). Then, we added a large amount of IN-CTD or IN-CTD-ΔCT (Appendix A, curve between 400 and 600 s). We did not observe any supplementary association of IN to TAR upon Tat binding. Overall, these results suggest that there is no competition between IN and Tat for TAR once Tat is bound first, nor a facilitating effect of Tat for IN binding to TAR. Therefore, this supports our hypothesis that IN binding to TAR precedes and facilitates Tat interaction.

### 2.4. Tat Competes with IN-CTD and Displaces It from TAR

To investigate the mechanism involved in Tat binding to the IN:TAR complex, we also performed a dose-dependent competition EMSA (Figure 4A); we added increasing concentrations of Tat to a preformed IN-CTD: or IN-CTD-ΔCT:TAR complex. Interestingly, we observed a dose-dependent interference between Tat and IN-CTD for TAR binding as demonstrated by the reappearance of free RNA at intermediate concentrations of Tat, suggesting that the presence of Tat displaces IN from the TAR RNA (Figure 4A, lanes 4 and 5). This is further supported by the fact that this interference disappeared at saturating concentrations of Tat, where displacement of IN would be complete and RNA uniquely bound to Tat (Figure 4A, lane 6). Interestingly, no interference was observed at any concentration of IN-CTD-ΔCT (Figure 4A, lanes 9 to 12), thus implying that the mechanism was dependent on the presence of the CT tail.

Since the migration pattern of the EMSA assay cannot discriminate between IN-bound and Tat-bound RNAs, we assessed whether the presence of Tat would modify the binding of IN to 3′ biotinylated TAR with pull-down assays (Figure 4B). IN-CTD and IN-CTD-ΔCT were efficiently co-precipitated (Figure 4B, lanes 1 and 5, respectively), whereas the addition of increasing concentrations of Tat serially decreased their binding to TAR (Figure 4B, lanes 2 to 4 and 6 to 8, respectively), confirming our hypothesis of a displacement of IN by Tat for TAR binding. Control experiments showed that Tat also interacted with TAR and was not precipitated when biotinylated TAR was absent (Figure 4B, lanes 9 and 10). Very small differences were detected between IN-CTD and IN-CTD-ΔCT during the TAR-pulldown assay (Figure 4B) compared to the EMSA assay (Figure 4A). This is likely due to the higher sensitivity offered by radioactivity labelling in EMSA, since nanomolar concentrations of proteins and RNA were used. Altogether, these results suggest that IN facilitates the binding of Tat to TAR, especially in the presence of the CT region. This allows Tat to bind TAR more efficiently until ultimately displacing IN from TAR.

## 3. Discussion

The 18 residue-long C-terminal tail of HIV-1 IN ensures several functions essential for infectivity. A comprehensive understanding of its involvement in the various steps of the infectious cycle is still lacking due to limited structural information and to the pleiotropic effects caused by its mutations [23,26,59]. Here, we demonstrated that this intrinsically disordered region of IN acts as a specific sensor for the peculiar structure of the apical stem-loop of TAR RNA by directly interacting with its apical hexaloop. We probed and quantified IN–TAR interaction by EMSA, nuclease digestion, and real-time bio layer interferometry. Data analysis suggested that IN-CTD modified TAR conformation, leading to an enhanced binding of Tat, especially when the CT region is present. Moreover, we highlighted an interplay between Tat, IN, and TAR, where Tat is competing with IN-CTD for TAR binding and is able to destabilize preformed IN-CTD:TAR complexes.

Mutations of TAR, aimed at altering the peculiar structure of the apical portion, decreased the relative affinity of IN-CTD compared to IN-CTD-ΔCT, suggesting a role of CT in the recognition of this portion of the TAR structure (Figure 2). Importantly, full-length IN (FLAG-IN-FLm and His-IN-FL) presented the same trend of sensitivity for TAR structural mutants (Appendix A). The presence of CT does not seem to be associated with a selective specificity of IN for TAR in vitro, as IN-CTD could bind efficiently to other structured RNAs regardless of the presence of CT. In particular, IN-CTD interacts with polyA, DIS, and SD/Ψ RNA elements of HIV-1 5′UTR (Figure 1D). This is consistent with previous data in which IN interaction with all these gRNA elements showed K_D_ values ranging from 3 to 90 nM [8]. The higher affinity previously observed for the full-length IN [8] compared to IN-CTD measured in this work (Figure 1F and Appendix A) could be due either to the presence of additional binding sites in the 5′UTR or within the NTD and CCD domains of full-length IN [8]. Thus, the CT region can recognize a TAR RNA with the proper apical stem-loop conformation among possible structurally defective TAR conformers, but it is not involved in the discrimination between viral structured RNA regions. Interestingly, this behaviour of IN-CT reminds us of that of p6, the C-terminal domain of HIV-1 Pr55Gag [60]. The Pr55GagΔp6 mutant, deleted from the p6 domain, showed no RNA binding specificity compared to the full-length Pr55Gag, suggesting that the presence of this region is required for the specific binding of Pr55Gag to DIS RNA within the 5′UTR of HIV-1 genome [60].

Previous biochemical and structural studies have demonstrated that the unstructured Tat Arginine-rich motif (ARM) penetrates the TAR major groove, which is made with stem-bulge-stem-loop secondary structures and mostly interacts with the U-rich bulge and nearby double-stranded regions (Appendix A; [39,61,62,63]). Despite its absence in the CT of a motif comparable to the Arginine stretch of Tat, our results suggest that the unstructured IN-CT region also binds the TAR major groove, but it does so through the interaction with G34 and, to a lesser extent, G36 within the hexaloop (Figure 3A). This could explain the lower affinity of IN-CTD for TAR mutants (Figure 2) where the G34 and G36 positions are altered/affected. Interestingly, upon IN-CTD or IN-CTD-ΔCT binding to TAR RNA, we observed an increased exposure of G32 of the hexaloop (Figure 3A). This result is also supported by a very recent study published by Liu and co-workers [57], in which a crosslinking-coupled SHAPE assay was used to probe the modifications induced in the 5′UTR upon IN-FL binding [57]. In their experiments, G32 also presented a greater reactivity after the binding of the HIV-1 IN-FL protein [57]. Moreover, SHAPE results indicated that IN-FL binding increased G32 reactivity, increased backbone flexibility of the apical loop of TAR, and increased crosslinking with C39 [57]. This observed flexibility could explain the TAR breakage around C39 that we observed upon IN-CTD and IN-CTD-ΔCT binding (namely U38 and C42, Figure 3A and Appendix A).

A complex of IN-CTD:TAR has been modelled following the IN-CTD:INI1183-304 structure based on the fact that the same 6 residues were engaged for the binding of IN-CTD to both INI1 and TAR [8,56] (Appendix A). In this model, IN-CTD binds the minor groove of the 4-nt apical stem of TAR. Noteworthy, the last C-terminal modelled residue (D270), which immediately precedes the CT tail, is oriented towards the major groove of TAR, which is consistent with our experimental hypothesis [56] (Appendix A).

The interaction of IN with gRNA has been shown to be critical for a proper localization of vRNP inside the protective capsid core [8,27,28,57]. In this context, TAR-selection is exerted by the IN-CT of the mature protein (alone or together with NC); while in the Pr160Gag-Pol precursor, this could be an additional mechanism to that of the Pr55Gag protein interaction with the packaging signals (reviewed in [64]) to allow the selective recruitment and encapsidation of the HIV-1 gRNA in the viral core.

Our data is also fully supported by recent in vivo evidence of the implication of IN in proviral transcription at early times after integration and in a Tat-independent manner [35]. The authors of this work demonstrated that the mutation of four Lysine residues within the IN-CTD was responsible for the reduction of proviral transcription [35]. Interestingly, the same Lysine residues are involved in the binding of IN to the viral RNA [8,27].

Along the same line, our data indicates that IN interacts with the TAR RNA and that this interaction favours the interaction of Tat to TAR. Indeed, the analysis of real-time binding kinetics revealed that the affinity of Tat for TAR increased when the latter is complexed with IN-CTD or IN-CTD-ΔCT beforehand (Figure 3B–E). This binding behaviour and the cooperative kinetics seem specific for TAR RNA compared to generic RNAs (Figure 3C–E). Moreover, our experiments suggest a temporal directionality of the binding reaction; Tat can bind the IN-CTD/CTD-ΔCT:RNA complexes but IN-CTD/CTD-ΔCT cannot bind Tat:RNA (Appendix A). Therefore, based on our data, we propose a temporal sequence for the interplay between IN, Tat, and TAR RNA in the early events of proviral transcription from the LTR; first, IN-CTD binds TAR on the nascent RNA through its C-terminal tail by contacting the apical stem-loop and, in particular, the hexaloop. This interaction modifies the structure of TAR (Figure 3A), promoting Tat binding (Figure 3B,C), which finally displaces IN from the TAR RNA in order to boost transcription elongation (Figure 4). Noteworthy, the competition of the Tat ARM with IN for TAR binding has also been reported elsewhere [8].

## 4. Materials and Methods

### 4.1. Protein Expression and Purification

Construction of plasmid pET15b encoding N-terminal 6XHis tagged IN-CTD (aa 220-270) and IN-CTD-ΔCT (aa 220-288) were previously reported [33]. The IN-full length (IN-FL) gene from pNL4-3 was cloned in a pPROEX-HTa vector in a frame with a 6XHis tag at the N-terminus. All proteins were expressed in *Escherichia coli* BL21(DE3) Rosetta/pLysS strain (Novagen, Sigma Aldrich, Saint-Quentin-Fallavier, France). Cells were grown in an LB medium supplemented with 10% (*w*/*v*) glucose, and protein expression was induced at an optical density at 600 nm (OD600) of 0.6 with 1 mM IPTG (isopropyl-β-d-thiogalactopyranoside). Cells were incubated overnight at 18 °C under continuous shaking then pelleted. The cell pellets were resuspended in a lysis buffer composed of 50 mM Tris-HCl pH 8, 1 M NaCl, 20 mM imidazole, 0.1 mM EDTA, 2 mM β-mercaptoethanol, and 10% (*w*/*v*) glycerol. Exclusively for bacterial lysis, the buffer was freshly supplemented with 2M urea, 2 mM of Adenosine triphosphate (ATP), 5mM CHAPS (3-[(3-cholamidopropyl)dimethylammonio]-1-propanesulfonate), and 1 tablet of Protease Inhibitor Cocktail (ROCHE, cOmplete™). The preparation was sonicated for 120 s on ice, then the resulting lysate was subjected to centrifugation at 11,000× *g* for 1h. The recovered supernatant was then applied to a HisTrap^TM^ Fast Flow Crude column (Cytiva Europe GmbH, France) and purified by an AKTA pure system (Cytiva Europe GmbH, France). The sample was first abundantly washed with a lysis buffer containing 100 mM imidazole and 2M NaCl, then the protein was eluted using a gradient up to 500 mM imidazole in 10 column volumes. A second step of purification was carried out using a Superdex 75 10/300 GL column (Cytiva Europe GmbH, France) with an isocratic elution carried out with a storing buffer (50 mM HEPES pH 7.5, 1 M NaCl, 5 mM β-mercaptoethanol, and 5% glycerol). Mammalian FlagIN-FL (IN-FLm) was expressed in Baby Hamster Kidney suspension cells (BHK21-C13-2P, Sigma Aldrich, Saint-Quentin-Fallavier, France) using a vaccinia virus expression system as previously described [45]. IN-CT peptide (YGKQMAGDDCVASRQDED) and 101-residue long Tat protein of the primary isolate 133 of HIV-1 were chemically synthesized [65,66]. This Tat has been biochemically characterized and its full biological activity had been previously validated [67].

### 4.2. In Vitro RNA Synthesis, Purification, and Radiolabeling

We produced several RNAs as listed in Appendix A using partially double-stranded templates formed by the hybridization of the T7 promoter-containing DNA oligonucleotides listed in Appendix A and following the protocol detailed in [68]. Templates for polyA, DIS, and SD/Psi RNAs were produced with PCR using a pNL4-3 plasmid as a template and T7 promoter-containing primers. RNA was transcribed with kit MEGAshortscript™ T7 (Thermo Fisher Scientific, Illkirch-Graffenstaden, France) following the manufacturer’s instructions, de-phosphorylated using Alkaline Phosphatase (New England Biolabs, Evry, France), and purified with phenol:chloroform:isoamyl alcohol (25:24:1) extraction and ethanol precipitation. The 3′ biotinylated TAR and RNA 30-mer (Supplementry Appendix A) were chemically synthesized (Sigma Aldrich, Saint-Quentin-Fallavier, France). RNA (50 pmol) was radiolabelled at the 5′ end using 10 units of T4 polynucleotide kinase (New England Biolabs) mixed to 3 μL of γ^32^P-ATP (3000 Ci/mmol, 10 mCi/mL, Perkin Elmer, Villebon-sur-Yvette, France) for 1 h at 37 °C then purified on denaturing 10% (*w*/*v*) polyacrylamide gel (29:1) as previously described [68]. Before use in binding and structural studies, RNA was heated in a refolding buffer (20 mM HEPES pH 7.5, 0.2 M NaCl, 2 mM MgCl_2_, 2 mM DTT) for 3 min at 95 °C followed by 40 min of slow controlled cooling to room temperature and finally placed on ice.

### 4.3. Electrophoretic Mobility Shift Assays (EMSA)

The electrophoretic mobility shift assays (EMSA) were performed as described in [69]. Samples were prepared by mixing a radiolabelled RNA with increasing concentrations of proteins, as indicated, in a buffer containing 20 mM MES pH 6.0, 150 mM NaCl, 2 mM DTT, 2 mM MgCl_2_, 0.2 µg BSA, and 8% (*v*/*v*) PEG8000. The samples were incubated at 37 °C for 30 min before being resolved by native 6% polyacrylamide (19:1) gel electrophoresis in a 0.5x TAE (Tris acetate EDTA) buffer. Results were analysed by phosphorimaging using ImageQuant software. IN-FLm was incubated for 2 h at 4 °C with the RNA substrate. For the dose-dependent competition assay shown in Figure 4A, we used for all the samples a constant saturating IN-to-TAR concentrations with an IN-to-TAR ratio per sample exceeding about ten times their respective K_D_. Then, we added increasing concentrations of Tat.

### 4.4. RNA Structural Probing

Enzymatic treatments were performed in 10 µL of the reaction mix containing 0.5 pmol of 5′ radiolabeled RNA, 0.2° µg of yeast tRNA, 1 × Structure buffer (Thermo Fisher Scientific, Illkirch-Graffenstaden, France), and 0.01 U of RNase T1 (Thermo Fisher Scientific, Illkirch-Graffenstaden, France). Incubation was done at 37 °C for 5 min. Reactions were stopped by the addition of 40 µL of a quenching buffer composed of 10 mM HEPES pH 7.5, 1 mM EDTA, and 3% SDS. Partial alkaline hydrolysis was performed as follows; 10 µL of reaction mix containing 0.5 pmol of RNA, 1 µg of yeast tRNA, and 1× Alkaline Hydrolysis buffer were incubated at 95 °C for 12 min then quenched with 2× denaturing loading buffer and placed on ice. For RNA/protein complexes, 0.5 pmol RNA were previously incubated with a 36 µM protein (final concentration) at 37 °C for 30 min then treated with RNase T1. After quenching, samples were phenol extracted and ethanol precipitated. After recovering from precipitation, all samples were run on a 15% sequencing polyacrylamide gel in 0.5 × TBE. (Tris Borate EDTA). Results were analysed by phosphorimaging.

### 4.5. Pulldown Assay

The pulldown assays were conducted as described before [68]. Briefly, proteins were mixed in a binding buffer (40 mM HEPES pH 7.5, 2.5 mM MgCl_2_, 2 mM β-mercaptoethanol and 5% glycerol) adjusting the final NaCl concentration to 150 mM. Samples were complemented or not with 600 pmol of 3′ end–biotinylated TAR (Appendix A) in a final volume of 30 μL, and incubated 30 min at 37 °C. To the mix were added 8 μL of magnetic streptavidin beads (Dynabeads MyOne, Thermo Fischer Scientific, Illkirch-Graffenstaden, France), and the mix was further incubated for 1h at 4 °C in a gentle rotation. The resin was washed three times with 500 μL BB-200 (40 mM HEPES pH 7.5, 200 mM NaCl, 2.5 mM MgCl_2_, 2 mM β-mercaptoethanol and 10% glycerol) on ice and proteins were eluted with an SDS loading buffer and analysed on a polyacrylamide 16% (37.5:1) SDS-PAGE. For the Histidine pulldown assay, magnetic streptavidin resin was replaced with HisPur™ Ni-NTA Magnetic Beads (Thermo Fischer Scientific, Illkirch-Graffenstaden, France) and protein elution was done with a 0.5 M Imidazole containing buffer.

### 4.6. Bio-Layer Interferometry

For the Bio-Layer Interferometry (BLI) analysis, we used the BLItz platform (FortéBio). For the high throughput experiments such as shown in Figure 1F and Appendix A, we used the Octet RED96e System (FortéBio, Avantor, VWR, Fontenay-sous-Bois, France). All measurements were performed in an assay buffer composed of 50 mM HEPES pH 7.5, 200 mM NaCl, 2 mM β-mercaptoethanol, 5% glycerol, 2 mM MgCl_2_, 1 µM ZnSO_4_, and 0.5 M BSA. For the BLItz platform, we used Streptavidin (SA) Biosensors (FortéBio, Avantor, VWR, Fontenay-sous-Bois, France), and for the Octet RED96e, we used the Streptavidin (SAX) Biosensors (FortéBio, Avantor, VWR, Fontenay-sous-Bois, France) that were hydrated for 10 min in an assay buffer then plunged in a solution containing 1 µM 3′ end–biotinylated TAR RNA in an assay buffer with a 1 × RNAase inhibitor (RNA Secure, Invitrogen, Thermo Fischer Scientific, Illkirch-Graffenstaden, France) and 0.1 µM BSA for the RNA loading step. A wash was performed after RNA loading. In the BLItz experiment, the association of the protein to RNA was monitored in real-time for 200 s; the hydrated biosensor tip was placed in a 1.5 mL black assay tube containing 200 µL of the protein solution in an assay buffer as indicated. Afterward, a dissociation step or a second association with Tat was performed for 200 s. Biosensors were discarded after each measurement. All kinetic assays shown in Figure 1F and Appendix A were performed using the Octet RED96e system and carried out using black 96-well plates, and samples were diluted in a freshly prepared assay buffer and incubated at 37 °C with an orbital shake. Association and dissociation steps were as in the BLItz experiments. Each time, reference sensors and negative control sensors were included. Sensorgrams were exported and data analysis was performed with KaleidaGraph software (Synergy Software, Reading, PA, USA).

## 5. Conclusions

In this study, we have brought quantitative evidence to the role exerted by HIV-1 Integrase C-terminal domain (IN-CTD) as an RNA binding protein in the very first stages of proviral transcription. In particular, we have shown that full length IN, despite its intrinsic instability and independently of the post-transcriptional modifications (i.e., recombinantly expressed in *E. coli* or in mammalian cells), binds TAR RNA (Figure 1 and Appendix A). The C-terminal domain (IN-CTD) also binds TAR RNA and the structured parts of the HIV-1 5′UTR (Figure 1D) as the full length, indirectly proving that IN-CTD is indeed the RNA-binding domain. Moreover, the last 18 residues of IN-CTD, which are predicted to be unstructured, only slightly affect the affinity of IN-CTD for TAR (Figure 1F). On the other hand, this same tail (IN-CT) is able to sense the shape of TAR RNA, in particular the hexaloop, and to modify it by exposing specific nucleotides, namely G32, U38, and C41 (Figure 2) and by protecting G34 and G36, as it was seen in in vivo SHAPE experiments [52]. This modification is preparatory to Tat binding to the same TAR RNA. Interestingly, the apparent affinity of Tat for TAR was about 115 times lower in the presence of IN-CTD and about 35 times lower in the presence of IN-CTD-△CT. Moreover, Tat binding to IN/TAR is cooperative (Figure 3). We further proved Tat competition and its ability to displace IN-CTD from TAR by interferometry, EMSA, and pull-down assays.

Overall, our in vitro data complements the in vivo observations [34] and together converge to the working model schematized in Figure 5, where IN-CTD selectively binds the nascent TAR transcript hexaloop with its CT tail. This binding modifies the TAR structure, facilitating its interaction with Tat, then Tat displaces IN and allows the subsequent transactivation of provirus transcription (Figure 5 bottom panel). This process is dynamic and might also be modulated by the post-translational modifications of IN, by the interaction with cellular partners, and/or by the chromatin remodeling processes.

Noteworthy, the structure of the HIV-1 intasome during the strand transfer process revealed that the inner IN tetrameric core, which contacts both the viral and host DNA, is surrounded by twelve other subunits [64,65] that might be available for other functions [11]. The structure of IN after strand transfer is not yet available, thus it is not possible to predict whether the CTD is at disposal for transcription.

Finally, our data adds to studies demonstrating the pleiotropic role of IN in viral packaging (reviewed in [27,52]), integration (reviewed in [9]), and early transcription [34] via its protein–RNA interactions. The direct role of CT in TAR binding puts IN back centre stage as a therapeutic target and opens the way to the development of a new generation of therapeutics.

## Figures and Tables

**Figure 2 ijms-23-13742-f002:**
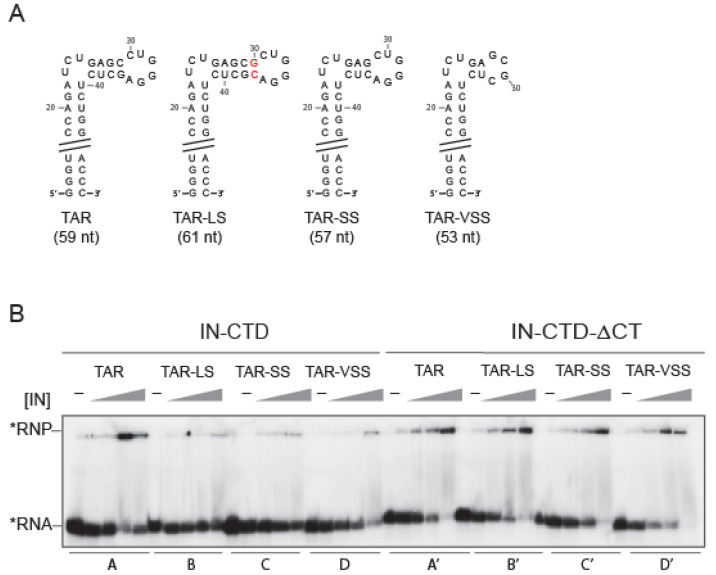
IN C-terminal tail sensing for TAR RNA apical stem-loop shape. (**A**) Structural model of TAR RNA and mutated versions used in our study: TAR long stem (TAR-LS, the red font underlines the added bases), TAR short stem (TAR-SS) and TAR very short stem (TAR-VSS). In order to verify that each RNA mutant assumed the expected secondary structure each model was analyzed with Fold WebServer [48,49]. (**B**) EMSA assay illustrating the interaction of IN-CTD and IN-CTD-ΔCT with TAR wild type and mutants. The RNA substrates (50 nM) are labelled with 32P (black star) and incubated with increasing concentrations of proteins (0; 100, 200, 400 and 800 nM) under the conditions described in Section 4.

**Figure 3 ijms-23-13742-f003:**
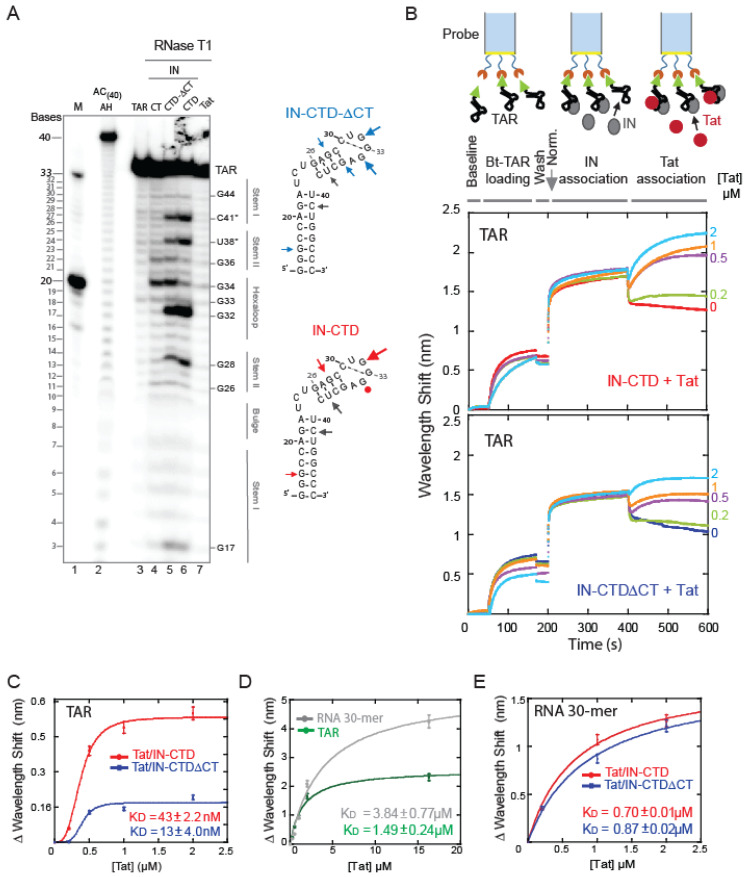
IN induces structural modifications of TAR, promoting Tat binding. (**A**) Probing the change in the secondary structure of TAR RNA complexed with different fragments of IN and Tat. 5′-end radiolabeled TAR (34-mer) was incubated in the presence or absence of protein for 30 min at 37 °C prior to RNase T1 treatment, as described in experimental procedures. RNA fragments were separated on a 15% denaturing polyacrylamide sequencing gel. Bands corresponding to certain T1 cleavage (at G bases) products are identified as position markers. Probing gel lanes are as follows: (M) Ladder of two RNA transcripts of 33 and 20 nucleotides in length (lane 1); (AC(40) AH) alkaline ladder of AC(40)-mer RNA (lane 2); TAR native (lane 3); TAR RNA complexes with IN-CT (lane 4); IN-CTD-ΔCT (lane 5); IN-CTD (lane 6) and Tat (lane 7). Digestion patterns were mapped on TAR secondary structure depicted on the right of the gel: circles identify nucleotides protected from RNAse T1 digestion and arrows mark the RNA cleavage sites. Grey arrows indicated nonspecific hydrolysis. The dimensions of arrows are proportional to the intensity of the band. (**B**) Real-time measurements of protein-RNA interaction by Interferometry (BLItz^®^ System instrument, FortéBio). The 3′ biotinylated TAR (Bt-TAR) was first loaded on the streptavidin-coated biosensor for 120 s (Bt-association) then the unbound RNA was washed for 30 s (wash). The sensor was absorbed in a solution containing about 90 μM of IN protein for 200 s then incubated with different concentrations of Tat (0.2, 0.5, 1 and 2 μM) for 200 s. (**C**) Analysis of interferometry shifts as a function of Tat concentration (from data shown in Panel B). Data points were fitted with Hill equation: y = Bmax ^xn^/K_D_ + (xn): K_D_ = 43 ± 2.2 nM (R^2^ = 0.99), n = 3 and Bmax = 0.57 ± 0.02 nm for Tat/IN-CTD (red curve) and K_D_ = 13 ± 4.0 nM (R^2^ = 0.96) n = 5 and Bmax = 0.18 ± 0.02 nm for Tat/IN-CTD-ΔCT (blue curve). (**D**) BLI experiment to measure Tat binding to immobilized Bt-TAR RNA (green line) or immobilized Bt-RNA(30)-mer (grey line). Data points were fitted with Michaelis-Menten equation. Tat affinities are: K_D_ = 1.49 ± 0.24 µM (R^2^ = 0.99) for TAR; K_D_ = 3.84 ± 0.77 µM (R^2^ = 0.99) for RNA(30)-mer. (**E**) BLI experiments as in B and C, but with immobilised RNA(30)-mer to test Tat ability to displace IN-CTD or IN-CTD-ΔCT (see methods section for protocol). Data points were fitted with Michaelis-Menten equation. K_D_ = 0.7 ± 0.1 µM (R^2^ = 0.99) for Tat/IN-CTD (red) and K_D_ = 0.87 ± 0.2 µM (R^2^ = 0.99) for Tat/IN-CTD-ΔCT (blue).

**Figure 4 ijms-23-13742-f004:**
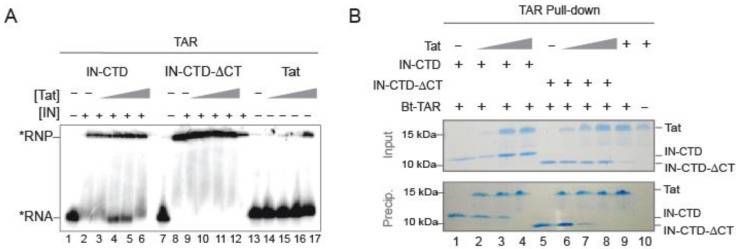
Tat competes against IN-CTD for TAR binding. (**A**) EMSA assay showing a dose-dependent competition assay of Tat on IN-CTD/TAR or IN-CTD-ΔCT/TAR complexes. TAR (50 nM), labelled with 32P (black star), was incubated with 16 fold molar excess of IN-CTD (800 nM, lanes 2–6) or IN-CTD-ΔCT (lanes 8–12) with or without increasing concentration of Tat (25, 50, 100, 200 nM). Tat vs IN-CTD is shown in lanes 3–6 and IN-CTD-ΔCT in lanes 9–12. Control binding of Tat on TAR was also performed using Tat alone (lanes 13–17). (**B**) Protein co-precipitation with 3′ end-biotinylated TAR (Bt-TAR). IN-CTD (lanes 1–4) or IN-CTD-ΔCT (lanes 5 and 8) were mixed with increasing amount of Tat (1, 2, 3 µg/sample; lanes 2, 3, 4 and 6, 7, 8) and incubated in a buffer containing 200 mM NaCl before co-precipitation. Tat was incubated with Bt-TAR alone (lane 9) or TAR-free beads as a control for unspecific binding (lane 10). Input (20% of total) and pull-down fractions were analyzed by 15% SDS-PAGE followed by Coomassie blue staining.

**Figure 5 ijms-23-13742-f005:**
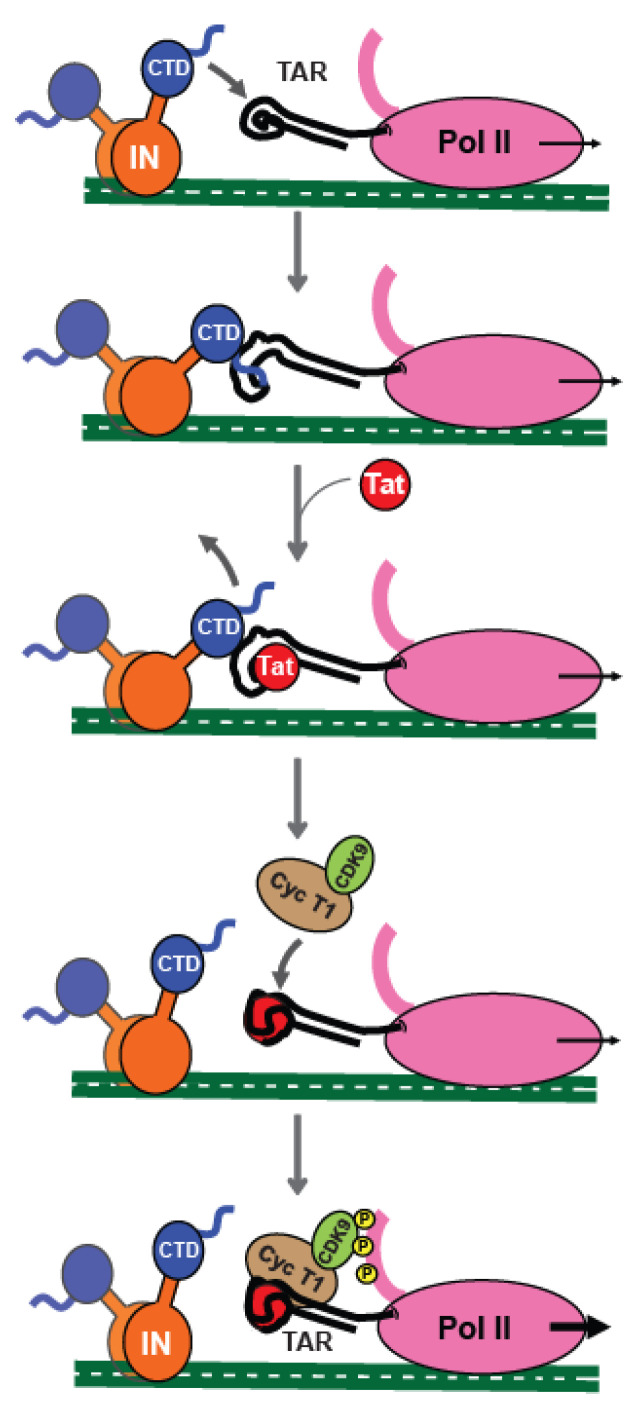
Cartoon of the working Model proposed for the interplay of IN-CTD, Tat and TAR during the early stage of HIV-1 proviral transcription. Once the integration has taken place, IN interacts with nascent TAR transcript through its CTD, directly interacting with the major groove and the hexaloop, thanks to its C-terminal tail. This binding induces TAR conformational changes, which promote the interaction of Tat with its substrate. Afterwards, Tat displaces IN-CTD from TAR and recruits the SEC complex in order to boost Pol II transcription.

## Data Availability

All additional data are available upon request.

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
