# Peer review of "The HIV-1 Integrase C-Terminal Domain Induces TAR RNA Structural Changes Promoting Tat Binding"

_ijms, 2022, doi:10.3390/ijms232213742_

Round 1

Reviewer 1 Report

The manuscript by Rocchi et al. showed that HIV-1 Integrase (IN) interacts with TAR RNA via the C-terminal domain (CTD) of IN. Also, they presented that the interactions of IN-CTD with TAR RNA promote Tat binding to TAR RNA. This manuscript includes several new findings which may lead to a critical discovery in the future, but publishing it in IJMS, it is required to perform multiple experiments.

Specific comments

1. Supplementary Figure 1A. The authors should provide images for IN-CTD and IN-CTD-ΔCT in addition to IN-FL because these recombinant proteins are essential tools for this study.

2. Figure 1C. IN-ΔCTD should be tested as a control to show CTD is a critical domain within IN for the interactions with TAR RNA.

3. Figure 2-4. Since it is unknown whether IN-CTD behaves as IN-FL, the authors should test IN-FL in addition to the mutants.

4. Figure 2B and Supplementary Figure 3A. IN-FL binds to TAR-LS, but IN-CTD does not. The authors should at least discuss this difference.

5. Lines 196-197. The results obtained by the experiments in Figures 1 and 2 could not suggest that IN-CT interacts with the apical stem loop of TAR. Do the authors want to say IN-CTD but not IN-CT?

6. Supplementary Figure 4E. There was no figure. The authors should provide the figure.

6. Lines 270-271 and Figure 4A. Tat binding to TAR RNA eliminates IN from TAR RNA. Is not this a kind of competition?

7. Lines 280-281 and Figure 4A. If Tat binding to TAR displaces IN, TAR RNA should form RNP with Tat. However, free RNA was observed. Does this suggest that the presence of Tat but not Tat binding to TAR eliminates IN from TAR RNA?

8. Lines 286-288 and Figure 4. Why the migration patterns of IN mutants-bound and Tat-bound TAR are similar even though the molecular weights of IN mutants and Tat are different? The authors should at least discuss this discrepancy.

9. Lines 293-297. The explanation of the differences in results obtained by the pull-down assay and EMSA is not clear.

10. Line 417. The results obtained by this study can not explain the direct interactions between the intrinsically disordered region of IN and TAR apical hexaloop.

Author Response

We would like to thank the Referee 1 for his/her remarkable work and for providing useful and detailed comments on an earlier version of the manuscript. He/She raised very interesting points that gave us the opportunity to better point out some questions and clarify some technical choices and make improve the manuscript.

Reviewer 1.

The manuscript by Rocchi et al. showed that HIV-1 Integrase (IN) interacts with TAR RNA via the C-terminal domain (CTD) of IN. Also, they presented that the interactions of IN-CTD with TAR RNA promote Tat binding to TAR RNA. This manuscript includes several new findings which may lead to a critical discovery in the future, but publishing it in IJMS, it is required to perform multiple experiments.

Specific comments

  1. Supplementary Figure 1A. The authors should provide images for IN-CTD and IN-CTD-ΔCT in addition to IN-FL because these recombinant proteins are essential tools for this study.

We thank the Referee 1 and we added a SDS PAGE of IN-CTD and IN-CTD-ΔCT in Supplementary Figure S1A.

  1. Figure 1C. IN-ΔCTD should be tested as a control to show CTD is a critical domain within IN for the interactions with TAR RNA.

The critical role played by the IN-CTD in RNA interaction has already been observed in vivo, in virio and in vitro (Kessl et al., 2016; Elliott et al 2020; Shema-Mugisha et al., 2022).

In particular, Kessl and co-workers reported that most of HIV-1 IN’s RNA-binding residues are present in the IN-CTD and the strongest affinity of HIV-1 IN, among all the structural elements of the HIV-1 5’UTR, is showed for TAR RNA. Moreover, mutation of basic residues within IN-CTD (i.e., R262, R263, R269, and K273) inhibits specifically the IN:RNA binding without altering IN oligomerization in virions and in vitro (Kessl et al., 2016 ; Elliot et al., 2020), suggesting that these residues directly mediate IN binding to the genomic RNA. Thanks to the referee’s comment, we understood that it was important to stress this point in the manuscript. So we cited the recent work of Shema-Mugisha et al., 2022 in the introduction of the manuscript (Now Ref. 34). In fact, Shema-Mugisha and co-workers isolated compensatory IN mutant viruses bearing substitutions of previously identified RNA-binding residues within the IN-CTD. These compensatory substitutions, present within the CTD and not elsewhere within the protein, restored the ability of IN to bind gRNA, highlighting the importance of the IN-CTD, and only the CTD, in mediating specific and direct RNA binding (Shema-Mugisha et al., 2022).

From a more technical point of view, the IN catalytic core domain, and the N-terminal region possesses very strong nucleic acid binding properties for the concerted integration enzymatic activity that could give biased RNA binding properties in in vitro experiment.

In conclusion we believe that performing experiments with IN-ΔCTD is out of the scope of our work, also considering the large amount of data presented on the IN-CTD from other labs.

  1. Figure 2-4. Since it is unknown whether IN-CTD behaves as IN-FL, the authors should test IN-FL in addition to the mutants.

As we mentioned in point 2, our objective was the study the IN-CTD, its interaction with TAR and the interplay with Tat. We used IN-FL only for preliminary set up and as a control. We consider that this request is out of the scope of the present manuscript and it is the object of a next manuscript to address other questions.

  1. Figure 2B and Supplementary Figure 3A. IN-FL binds to TAR-LS, but IN-CTD does not. The authors should at least discuss this difference.

As we explained in 2.1 session the IN-FL is poorly soluble in vitro and people overcome this problem by mutations of hydrophobic residues. These mutations results in a replication-defective virus with mislocalized viral RNP phenotype, analogous to that observed in IN mutants defective for the IN-gRNA interactions.

In this work we really wanted to get as close as possible to physiological conditions in terms of ionic strength and for this reason the full length IN was not suitable. We added the Supplementary Figure S3A only to show an analogous trend of binding of IN-FL to TAR mutants compared to what observed for IN-CTD. The interesting observation is still the difference of IN-CTD and IN-CTD-ΔCT binding to TAR mutants that is showed in main Figure 2B.

  1. Lines 196-197. The results obtained by the experiments in Figures 1 and 2 could not suggest that IN-CT interacts with the apical stem loop of TAR. Do the authors want to say IN-CTD but not IN-CT?

Thank you, it was indeed a typo,  we changed IN-CT with IN-CTD in lane 196

  1. Supplementary Figure 4E. There was no figure. The authors should provide the figure.

We apologize, we made a mistake: the figure concerned was Supplementary Figure 5D. We changed from 4E to 5D in line 264.

  1. Lines 270-271 and Figure 4A. Tat binding to TAR RNA eliminates IN from TAR RNA. Is not this a kind of competition?

Concerning Figure 4A Referee 1 is right: this is a kind of completion. However we preferred to talk about “interference” because the EMSA assay (Figure 4.A) cannot allow the discrimination between IN:TAR and Tat:TAR complexes and in the case of IN-CTD both proteins detached from TAR (Figure 4A lanes 4-5). The competition is visible in Figure 4B where Tat replaced IN-CTD or IN-CTD- ΔCT on TAR RNA in pull-down assay.

Lines 270-271 are referred to the other way around experiment showed in Supplementary Figure 5B where we first incubated Tat with TAR then we added IN-CTD/IN-CTD- ΔCT. In this case any supplementary binding or competition are observed. However, for a better understanding of our purposes we changed: lanes 270-271 that now states “Overall, these results suggest that there is no competition between IN and Tat for TAR, once Tat is bound first, nor a facilitating effect of Tat for IN binding to TAR.”

  1. Lines 280-281 and Figure 4A. If Tat binding to TAR displaces IN, TAR RNA should form RNP with Tat. However, free RNA was observed. Does this suggest that the presence of Tat but not Tat binding to TAR eliminates IN from TAR RNA?

This is a good point, the referee 1 is right. In this window of concentration of Tat, neither IN-CTD nor Tat bind TAR (Figure 4A lanes 4-5). We rephrased lanes 280-281 that now state “Interestingly, we observed a dose-dependent interference between Tat and IN-CTD for TAR binding as demonstrated by the reappearance of free RNA at intermediate concentrations of Tat, suggesting that the presence of Tat displaces IN from the TAR RNA (Fig. 4A, lanes 4 and 5).” 

  1. Lines 286-288 and Figure 4. Why the migration patterns of IN mutants-bound and Tat-bound TAR are similar even though the molecular weights of IN mutants and Tat are different? The authors should at least discuss this discrepancy.

EMSA is a native gel where proteins or protein/nucleic acid complexes are separated according to their net charge and size. We observed an entry of the protein/RNA complex of about 1 or 2 millimeters from the bottom of the well, most likely because the complex possess a low net charge to migrate more. Our IN-CTD/IN-CTD- ΔCT/Tat :TAR complexes migrated too high in the gel and we do not have enough resolution to discriminate them. Moreover the difference in molecular weight between Tat and IN-CTD is too small (about 5 kDa) to be eventually visible in a 6% polyacrylamide gel.

  1. Lines 293-297. The explanation of the differences in results obtained by the pull-down assay and EMSA is not clear.

We rephrased lines 293-297 that now states: “Very small differences were detected between IN-CTD and IN-CTD-ΔCT during TAR-pulldown assay (Fig. 4B) compared to EMSA assay (Fig. 4A). This is likely due to the higher sensitivity offered by radioactivity labelling in EMSA, since nanomolar concentrations of proteins and RNA were used.”

  1. Line 417. The results obtained by this study cannot explain the direct interactions between the intrinsically disordered region of IN and TAR apical hexaloop.

We actually showed the interaction of the IN CT tail with the apical hexaloop of TAR. In fact, in the footprinting assay showed in Figure 3A we observed two different digestion patterns for IN-CTD and IN-CTD-ΔCT. IN-CTD showed a protection of G34 and, in a lesser extent G36 that are localized in the apical hexaloop.

Moreover, We noticed that in the scheme on the right of Figure 3A the number 33 was displaced from its original position, we corrected this mistake.

Reviewer 2 Report

Authors in their paper analyzed in details integration among TAR RNA, HIV-1 Tat protein and the C-terminus of HIV-1 Integrase. They provide a model regarding interplay of these proteins in early stages of HIV infection.

Experiments are well designed, methods are described in sufficient details and all results are adequately discussed. All major experiments are back-uped with at least 2 methods which supports the conclusions. I have no major objections.

Minor points/comments:

1) Fig. 1F: -using BLI, authors observed small difference between binding of IN-CTD and IN-CTD_ΔCT to TAR RNA. I am not an expert working with BLI, so maybe this comment is irrelevant, but could it be possible that such a small difference is caused just by the difference in protein size (for example by altering diffusion rate of proteins)? Would it be better to immobilize the proteins and use TAR RNA as a binding partner – it would be the same in both cases?

2)  Fig. 4A (but also Fig. 2B, supplementary Fig. 3): quantification of activity on EMSA gels (and resulting protein-RNA binding quantification) is calculated by using both values of shifted (bound) and free (unbound) probe. However, such a calculation could be misleading – I would prefer to rely just on the shifted band/activity as this directly corresponds to the amount of bound probe. Probe is usually used in excess and overall amount of probe could influence dynamic of binding to protein… Especially conclusions based on Fig. 4A, lanes 4 and 5 (text lines 279-281) and supplementary Fig. 3B (lines 194-195) are not much convincing and personally, I do not see claimed changes on the bound signal. Did authors have some specific reasons to calculate data in the presented form and not just using bound probe signal (which I would prefer)?

3)  Authors claimed that EMSA experiments (Fig. 4A) cannot distinguish Tat and IN binding to TAR probe, but this could be done using supershift with antibody – as IN protein are tagged. Did authors try such experiments? It would confirm the conclusions and disprove doubts (lines 294-295).

4) Highlight panel labels (A,B,C…) in figure legends (at least similarly to supplementary figures) – it will improve orientation in text.

5) Citation of supplementary Fig. 4E (line 1264) should be rather 5D?

6)  Is Hill equation written correctly in line 377? It is not much clear.

7) Typos: parentheses (lines 52, 56, 57, 60, 181, 466, 507), spaces (lines 237, 239, 257, 318, 460), IN-CTD (lines 62, 185), represent (line 166), under our conditions (line 205), Fig. 3E (line 258), correct bold font (Fig. 4B citations; lines 288-295), RNase (line 369), IN-CTD-CT (line 384), symbol “µ” (lines 382-386), delete “with” (line 549)

Author Response

We would like to thank Referee 2 for having carefully read our manuscript and for giving such encouraging comments. He/She raised very interesting point that we have addressed either in this answer or in the main text.

Reviewer 2.

Authors in their paper analyzed in details integration among TAR RNA, HIV-1 Tat protein and the C-terminus of HIV-1 Integrase. They provide a model regarding interplay of these proteins in early stages of HIV infection.

Experiments are well designed, methods are described in sufficient details and all results are adequately discussed. All major experiments are back-uped with at least 2 methods which supports the conclusions. I have no major objections.

Minor points/comments:

1) Fig. 1F: -using BLI, authors observed small difference between binding of IN-CTD and IN-CTD_ΔCT to TAR RNA. I am not an expert working with BLI, so maybe this comment is irrelevant, but could it be possible that such a small difference is caused just by the difference in protein size (for example by altering diffusion rate of proteins)? Would it be better to immobilize the proteins and use TAR RNA as a binding partner – it would be the same in both cases?

We thank the refree for raising this good point; it is true that a small mass effect could be possible. However, the small differences observed for IN-CTD and IN-CTD-ΔCT have no impact because we extrapolated the relative Δshift of the wavelength for the calculation of the apparent KDs. 

Concerning the experiment proposed: in preliminary tests we immobilized the IN-CT and IN-CT-DCT proteins on reactive amine biosensors, then we soaked it in TAR solution to monitor the binding. This immobilization gives a crowded coating and also the orientation is not the same, since it is based on random Lys immobilization. Thus the results obtained in this way showed a high background noise. The use of streptavidin-coated biosensor to immobilize the biotinylated RNA gave us much more clean and reproducible experiments, most likely because the TAR RNA was homogeneously sparse on the sensor and the right binding side always exposed for binding.

2)  Fig. 4A (but also Fig. 2B, supplementary Fig. 3): quantification of activity on EMSA gels (and resulting protein-RNA binding quantification) is calculated by using both values of shifted (bound) and free (unbound) probe. However, such a calculation could be misleading – I would prefer to rely just on the shifted band/activity as this directly corresponds to the amount of bound probe. Probe is usually used in excess and overall amount of probe could influence dynamic of binding to protein…Especially conclusions based on Fig. 4A, lanes 4 and 5 (text lines 279-281) and supplementary Fig. 3B (lines 194-195) are not much convincing and personally, I do not see claimed changes on the bound signal. Did authors have some specific reasons to calculate data in the presented form and not just using bound probe signal (which I would prefer)?

We understand this point. We used EMSA assays only for qualitative purposes while, to have a more quantitative analysis and to follow the binding in real time, we used interferometry experiments. The only exception was for Fig. 2B: therefore we added in Fig. 2C the quantification of four independent EMSAs. In this case, the quantification seems important for better visualizing the effect of the presence of CT tail in the binding of TAR mutants. To better quantify the bands, we took care of performing this EMSA on the same gel, testing the two proteins each one on four TAR mutants. Despite this careful setup, we agreed with Referee 2 and we did not find correct to extrapolate statistical information from this analysis. Therefore,  we have decided to remove the panel C from Figure 2 representing the quantification of the EMSA presented in Figure 2B and panel C of Supplementary Figure S3. The text is modified accordingly. 

3)  Authors claimed that EMSA experiments (Fig. 4A) cannot distinguish Tat and IN binding to TAR probe, but this could be done using supershift with antibody – as IN protein are tagged. Did authors try such experiments? It would confirm the conclusions and disprove doubts (lines 294-295).

This is a good idea. However, we observed an entry of the protein/RNA complexes of about 1 or 2 millimeters from the bottom of the well, most likely because the IN-CTD/IN-CTD- ΔCT/Tat :TAR complexes possess a low net charge to migrate more. Unfortunately, this band migrated too high in the gel to perform a super shift.

4) Highlight panel labels (A,B,C…) in figure legends (at least similarly to supplementary figures) – it will improve orientation in text.

Thank you for the comment, we changed as required.

5) Citation of supplementary Fig. 4E (line 1264) should be rather 5D?

Indeed, it was a typo. Thank you, we have corrected it.

6)  Is Hill equation written correctly in line 377? It is not much clear.

We apologize for the mistake, the superscript format was lost during formatting: we corrected with y=Bmax xn /(KD+ xn).

7) Typos: parentheses (lines 52, 56, 57, 60, 181, 466, 507), spaces (lines 237, 239, 257, 318, 460), IN-CTD (lines 62, 185), represent (line 166), under our conditions (line 205), Fig. 3E (line 258), correct bold font (Fig. 4B citations; lines 288-295), RNase (line 369), IN-CTD-CT (line 384), symbol “µ” (lines 382-386), delete “with” (line 549).

Thank you for these comments, we have corrected them all.

Round 2

Reviewer 1 Report

The authors have addressed all my comments and all revisions made have substantially improved the manuscript’s clarity. The study provides novel findings, which are interesting to the readership of IJMS. I recommend the publication of the manuscript.